# Bidirectional Functional Effects of *Staphylococcus* on Carcinogenesis

**DOI:** 10.3390/microorganisms10122353

**Published:** 2022-11-28

**Authors:** Yuannan Wei, Esha Sandhu, Xi Yang, Jie Yang, Yuanyuan Ren, Xingjie Gao

**Affiliations:** 1Faculty of Science, University of Manitoba, Winnipeg, MB R3T 2N2, Canada; 2Department of Immunology, University of Manitoba, Winnipeg, MB R3T 2N2, Canada; 3Department of Biochemistry and Molecular Biology, School of Basic Medical Science, Tianjin Medical University, Qixiangtai Road No. 22, Heping District, Tianjin 300070, China; 4Department of Immunology, School of Basic Medical Science, Tianjin Medical University, Qixiangtai Road No. 22, Heping District, Tianjin 300070, China; 5Key Laboratory of Immune Microenvironment and Disease (Ministry of Education), Key Laboratory of Cellular and Molecular Immunology in Tianjin, Excellent Talent Project, The Province and Ministry Co-sponsored Collaborative Innovation Center for Medical Epigenetics, Tianjin Medical University, Qixiangtai Road No. 22, Heping District, Tianjin 300070, China

**Keywords:** *Staphylococcus*, cancer, *S. aureus*, staphylococcal nuclease, SND1

## Abstract

As a Gram-positive cocci existing in nature, *Staphylococcus* has a variety of species, such as *Staphylococcus aureus* and *Staphylococcus epidermidis*, etc. Growing evidence reveals that *Staphylococcus* is closely related to the occurrence and development of various cancers. On the one hand, cancer patients are more likely to suffer from bacterial infection and antibiotic-resistant strain infection compared to healthy controls. On the other hand, there exists an association between staphylococcal infection and carcinogenesis. *Staphylococcus* often plays a pathogenic role and evades the host immune system through surface adhesion molecules, α-hemolysin, PVL (Panton-Valentine leukocidin), SEs (staphylococcal enterotoxins), SpA (staphylococcal protein A), TSST-1 (Toxic shock syndrom toxin-1) and other factors. Staphylococcal nucleases (SNases) are extracellular nucleases that serve as genomic markers for *Staphylococcus aureus*. Interestingly, a human homologue of SNases, SND1 (staphylococcal nuclease and Tudor domain-containing 1), has been recognized as an oncoprotein. This review is the first to summarize the reported basic and clinical evidence on staphylococci and neoplasms. Investigations on the correlation between *Staphylococcus* and the occurrence, development, diagnosis and treatment of breast, skin, oral, colon and other cancers, are made from the perspectives of various virulence factors and SND1.

## 1. Introduction

*Staphylococcus* is a group of Gram-positive cocci that contains many different species, such as *Staphylococcus aureus* (*S. aureus*), *Staphylococcus epidermidis* (*S. epidermidis*), *Staphylococcus saprophytics* (*S. saprophytics*) [1,2,3,4]. As the most common pathogenic bacteria, *S. aureus* with different sequence types (STs) or *spa* types can cause inflammatory reactions in humans and animals [1,4,5]. The *S. aureus*-induced community and hospital-acquired infections may lead to adverse effects on the treatment and prognosis of patients [4]. With the widespread use of antibiotics in clinical practice, *S. aureus* has gradually become more drug-resistant, and the detection rate of methicillin-resistant *Staphylococcus aureus* (MRSA) also shows an upward trend [6]. Interestingly, *Staphylococcus lugdunensis* (*S. lugdunensis*) can secrete a polypeptide antibiotic called lugdunin to effectively restrain reproduction of and infection with MRSA [7]. As one of the main microorganisms on the skin’s surface, S. epidermidis plays an important role in the epidermal defense system of the body [8]. At present, more and more evidence supports the functional correlation between *Staphylococcus* and tumors, which is discussed in this review.

*Staphylococcus* in the host can play the role of inducing pathogenicity and escape from the host immune system through a variety of virulence factors, such as surface adhesion molecules, exotoxins and exoenzymes [9,10]. Various cell wall protein-anchored surface proteins, such as fibronectin-binding protein A/B (FnBPA/B), contribute to the adherence of *Staphylococcus* to host cells, which is the key to the staphylococcal pathogenesis [10,11,12,13]. As poreforming bacterial toxins, alpha-hemolysin and Panton-Valentine leukocidin (PVL) are considered to be the main virulence factors of severe infection caused by *S. aureus* infection [4,9,14]. A series of staphylococcal superantigens (SAg) produced by *S. aureus* can effectively activate the proliferation of T and B cells without any processing by antigen-presenting cells [9,15,16]. SpA (staphylococcal protein A) is one of the most important cell wall proteins in *S. aureus*, and has B cell superantigen activity [9]. SEs (staphylococcal enterotoxins) and TSST-1 (toxic shock syndrom toxin-1) function as potent inducers of cytotoxic T lymphocyte activity and cytokine production [15,16]. SEs include the *Staphylococcus aureus* enterotoxin A/B/C (SEA/B/C), and SEC is further divided into three subtypes (C1/2/3) [17,18]. TSST-1 can lead to toxic shock syndrome, and even multiple organ failure [19].

Extracellular nuclease is a secreted virulence factor and genetic marker for *S. aureus*. There exist two types of extracellular nuclease, staphylococcal nucleases (SNases) and thermonucleases (TNases) [20,21,22]. SND1 (staphylococcal nuclease and Tudor domain-containing 1) is the human homologue of *Staphylococcus aureus* nuclease, and can work as a member of RNA-induced silencing complex (RISC) that takes part in the cleavage of mRNA [23,24,25]. It is currently believed that human SND1 consists of four repeating staphylococcal nuclease-like (SN-like) domains [SN(1–4)] at the N terminus, and a SN5a-Tudor-SN5b (TSN) domain at the C terminus [25,26,27]. SND1 is a multifunctional protein that plays an important role in gene transcription regulation, pre-mRNA splicing, cell cycle, RNA metabolism and other biological processes [25,26,28,29,30,31,32,33]. Furthermore, a growing body of evidence reveals that SND1 with a recognizable nuclease domain is a kind of oncoprotein closely related to the occurrence and development of tumors, and which involves the potential nuclease activity [25,34,35,36,37].

In this study, we first conducted a retrieval from the Pubmed database using the search term: “(((((((((((*Staphylococcus*) or (*Staphylococcus aureus*)) or (*Staphylococcus epidermidis*)) OR (*Staphylococcus saprophytics*)) or (*S. aureus*)) or (*S. epidermidis*)) or (*S. saprophytics*)) or (*Staphylococcus lugdunensis*)) or (*S. lugdunensis*)) or (SND1)) or (staphylococcal nuclease)) AND ((((((carcinogenesis) or (cancer)) OR (cancers)) or (tumor)) or (tumors)) or (tumorigenesis))”. Then, the obtained literature was screened by reading the abstracts or full texts. Finally, we selected a total of 78 articles containing the scientific data between the presence of *Staphylococcus* and the occurrence, development, and treatment of different types of cancer. Table 1 summarizes the relevant clinical reports and basic experimental evidence, in terms of surface adhesion molecules, α-hemolysin, PVL, SEs, TSST-1, SpA, and SND1.

## 2. *Staphylococcus* and Cancer-Related Clinical Reports

After the systematic literature research, a series of publications were retrieved regarding *Staphylococcus* and different clinical tumor diseases. For instance, when compared with negative controls, cancer patients tend to develop staphylococcal infections, and suffer from MRSA, which also greatly reduces the survival rate of patients with malignant tumors [40,61,86,91,109,113,116,117]. A 3-year retrospective study from a comprehensive cancer center reported that *S. lugdunensis* causes infection much less often than other coagulase-negative staphylococci species [81]. On the other hand, *S. aureus* is frequently detected in the oral cavity of most patients with malignant tumors undergoing chemotherapy and/or radiotherapy [47,58,86,87]. Maślak, E. et al. also observed the changes of *Staphylococcus* in the urine sample of prostate cancer patients treated with radiotherapy [112]. A study of an *S. aureus* bacteremia (SAB) case in a national database (*n* = 12,918) and a random population cohort (*n* = 117,465) analyzed the risk of primary cancer and discovered that SAB cases appeared more frequently in multiple myeloma, leukemia, sarcoma, cervical, liver, pancreatic, and urinary tract cancer, compared with a control group [100].

Microbiome sequencing and functional analysis for tumor and non-tumor patients will help to explore the correlation between staphylococcal system disorders and tumorigenesis prevention or treatment. Herein, we have gathered the scientific data on the functional relationship between staphylococci and several types of cancers.

### 2.1. Breast Cancer

Emerging evidence supports the links of *Staphylococcus* with breast diseases, especially breast cancer [99,118]. There are many clinical cases of breast cancer with MRSA [44]. *Staphylococcus* exhibits distinct distribution characteristics in different pathological tissues or states. For example, a relative abundance of *Staphylococcus* was detected in the breast tissues of women with breast cancer [78,82,97,98]. For instance, as the second most dominant bacterium, *Staphylococcus* (6.4% ± 9.4%) was prevalent in 22 out of 23 breast tissue samples of cases within black or white non-Hispanic cohorts of breast cancer [97]. Additionally, *S. aureus* and *S. epidermidis* are the common bacteria that cause infections around breast implants in cancer patients [72]. However, there are also reports with inconsistent conclusions. Breast microbiome profile data showed that the presence of *Staphylococcus* is negligible in the tissue of breast cancer [107], but An, J. et al. reported that the blood sample of healthy controls had a greater diversity of *Staphylococcus* than breast cancer patients [111].

### 2.2. Skin Cancer

In contrast to healthy skin, the presence of *S. aureus* DNA was strongly associated with squamous cell carcinoma [52]. Madhusudhan, N. et al. further reported that excessive *S. aureus* is significantly associated with an increased expression of human β-defensin-2 (HBD-2) in tumor samples from patients with cutaneous squamous cell carcinoma [93]. Cutaneous colonization of *S. aureus* is reportedly associated with the incidence of cutaneous T-cell lymphoma [69,119]. In response to adverse external stimuli, the expression microbiome of the body may become disorganized, such potentially suffering from a reduced level of the anti-tumor *S. epidermidis* population or a higher abundance of pathogenic *S. aureus*, which is associated with a high susceptibility to skin cancer [88,120,121]. When tumor patients are given specific clinical treatments, such as radiotherapy, chemotherapy, and probiotics, disorders of the skin microbiome are often observed [120,121].

### 2.3. Bladder Cancer

The altered abundance of *Staphylococcus* was detected in the tumor mucosa or urine samples of bladder cancer patients. For instance, *Staphylococcus* (cluster 2) was enriched in the microbial composition of tumor mucosa samples for bladder cancer [108]. Urine microbiota analysis of male bladder cancer patients in China indicated that various functional pathways were enriched in the cancer group, including *S. aureus* infection [85]. An abundance of *Staphylococcus* was significantly higher in urine samples of bladder cancer patients compared to benign prostatic hyperplasia controls [114].

### 2.4. Colon Cancer

In 2007, Noguchi, N. et al. first reported that tannin-producing *S. ludunensis* was more frequent in the swab samples of fecal and rectal for the advanced colon cancer group compared with the adenoma or normal group [48]. Furthermore, the genetic background investigation of the forty *S. lugdunensis* isolates from 288 rectal swabs indicated the links between the specific group D clone of *S. lugdunensis* and colon cancer [89].

### 2.5. Oral Cancer

Compared with healthy individuals, *Staphylococcus* was significantly more abundant in the oral squamous cell carcinomas group [110]. In 2004, Fujiki H. et al. found that tobacco tar-resistant *S. aureus* exists in the oral cavity of some individuals and has carcinogenic potential [42]. In addition, a study of 186 patients with oral squamous cell carcinoma reported a predominance of Gram-positive bacteria, including *S. aureus* and *S. epidermidis*, in the mouth of patients treated with chemotherapy and chemoradiotherapy [58].

### 2.6. Others

Apart from working to induce the discussed cancers, there are links between *Staphylococcus* and lung cancer, glioblastoma, and lymphoma. Fourdrain, A. et al. reported that the *S. aureus* carried in the nasal cavity before lung cancer surgery is related to an increased risk of health care-associated infection [94]. Similarly, *S. epidermidis* can also be detected in tissue samples taken from lung cancer patients during surgery [57]. In some glioblastoma multiforme cases, intracranial abscess complications caused by *S. aureus* have been observed [51]. Interestingly, some glioblastoma patients with staphylococcal intracranial infection after craniotomy displayed a relatively longer survival time [105]. However, the results are conflicting in breast implant-associated anaplastic large cell lymphoma (BIA-ALCL). It was reported that there was a high abundance of *Staphylococcus* in both breast implant-associated anaplastic and contralateral breast controls [90], but Hu H. et al. reported a lower abundance of *Staphylococcus* in the BIA-ALCL samples compared to that in the nontumor capsule specimens [79].

## 3. Staphylococcal Nuclease and Cancer

The presence or absence of *S. aureus* in samples can be determined by their diagnostic marker, staphylococcal nucleases [122]. Nucleases have long been recognized as potential biomarkers of cancer [36], however, no direct correlation between staphylococcal nucleases and cancer has been reported. The staphylococcal nuclease is a small globular protein containing 149 amino acid residues, and has been utilized to study the protein folding process [123]. As the staphylococcal nuclease purifies from a recombinant *E. coli* strain, micrococcal nuclease (Mnase) was applied in the chromatin immunoprecipitation assay or single-cell micrococcal nuclease sequencing of tumor samples [124,125]. SND1 is a conformed oncoprotein [25,34,35], which is the human homologue of SNases and contains four staphylococcal nuclease-like domains [23,24].

### 3.1. Structural Characteristics

Human SND1 protein (NP_055205.2; A0A140VK49_HUMAN), coded by the SND1 gene localized on chromosome 7q32.1 [34,126,127], consists of 910 amino acids. In 1997, Callebaut I. et al. first utilized the hydrophobic cluster analysis (HCA) method to initially resolve the structure of human SND1 protein and found that SND1 consists of four repetitive N-terminal SN and C-terminal Tudor domains [128]. In 2007, we first resolved the crystal structure of the TSN domain in human SND1 protein and found that TSN contains four α-helices, nine β-folds, and 14 linkage loops, in which the β (1~2) fold is involved in the composition of SN5a (679–703) [26]. Most of the α1-helices and β (3~6) fold to form a typical β-barrel Tudor (704–793) domain, and the β (7–9)-fold and α (2–4) helix are involved in the composition of SN5b (794–895) [26]. In 2008, Li, C. L. further reported that the SN3, SN4, Tudor and SN5 domains of human SND1 protein aggregate together to form a crescent-like structure [27]. The recessed basic surface formed by SN3 and SN4 serves as a binding site for citrate ions at the RNase active site, which can specifically bind with and degrade highly edited IU- and UI-containing double-stranded microRNA precursors [27]. Thus, staphylococcal nuclease-like domains of SND1 can bind to proteins and nucleic acids. This may involve a synergistic interaction between multiple SN structures.

### 3.2. Staphylococcal Nuclease Activity

The staphylococcal nuclease (SN) is a type of Ca^2+^-dependent enzyme that hydrolyzes the 5′-phosphodiester bond of single/double-stranded DNA and RNA [129,130]. It was initially thought that the SN domains of SND1 proteins lack key catalytic residues, like those of staphylococcal nucleases [24,128]. It was speculated that SND1 might have only nucleic acid binding ability, but no nuclease activities.

Nevertheless, emerging evidence suggests that the SND1 protein in multiple species can bind nucleic acids [27,131,132,133,134,135,136] and exhibits some nuclease activity [23,27,131,137,138,139,140,141,142,143,144]. For instance, Hannon et al. first discovered that the SND1 is a candidate of RISC and shows the nuclease activity in mammalian, Drosophila, and Caenorhabditis elegans, despite lacing a classical active site sequence [23,137]. In Plasmodium falciparum, the SND1 protein can degrade the RNA and single-stranded DNA, displaying Ca^2+^-dependent nuclease activity [131]. The nuclease activity of the SND1 protein was also detected in the species of Tick, Penaeus monodon, and Toxoplasma gondii [140,142,143,144]. In addition, the SND1 protein has some degradation ability for pri-miRNA/dsRNA and specific types of miRNAs after RNA editing which is supported by the crystal structure evidence [27]. SND1 protein degrades highly edited A to I pri-miR-142 [138]. Additionally, SND1 also specifically binds and degrades I-dsRNAs enriched in IU base pairs, without interacting with IU base pair-free dsRNAs [139].

### 3.3. SND1 and Cancer

The potential nuclease activity of the SN domain within SND1 may be closely linked to the oncogenic role of the SND1 protein [25,34,35,36,37]. SND1 plays a vital role in regulating several aspects of RNA metabolism through its nuclease activity. For instance, the binding of SND1 to the 3′UTR of PTPN23 (protein tyrosine phosphatase nonreceptor type 23) mRNA in human hepatocellular carcinoma (HCC) promotes its RNA degradation [37]. As a conventional staphylococcal nuclease inhibitor, pdTp (3′,5′-deoxythymidine bisphosphate) was reported to suppress the nuclease activity of SND1 [131,137]. In HCC cells, the remarkably enriched RISC activity of SND1 depends on the nuclease activity of highly expressed SND1, which can be affected by pdTp [56]. For the subcutaneous or in situ mouse models of HCC, the treatment of pdTp injection hinders the tumorigenesis of mice by affecting the nuclease activity of SND1 [84]. Scholarship generally concludes that the inhibition of SND1 nuclease activity by pdTp could be an effective intervention or therapeutic strategy for hepatocellular carcinoma.

## 4. *Staphylococcus* and Cancer Treatment

Clinical evidence indicates a correlation between the occurrence, development, and treatment of cancer and *Staphylococcus* [145]. In many cases, the predisposition to tumors is accompanied and facilitated by infection with specific staphylococci. Hattar, K. et al. reported that lipoteichoic acid, an inflammatory mediator from *S. aureus*, promotes the proliferation of lung cancer cell lines (A549 and H226) in vitro [83]. *S. aureus* infection was found to promote the lung metastasis of breast cancer cells through the formation of neutrophil extracellular traps [101]. Hence, some tumor-related interventions can be conducted, partly based on the pathogenesis of *Staphylococcus*. For instance, it may be possible to evade drug resistance in *Staphylococcus* and tumors by regulating intracellular reactive oxygen species [146].

Interestingly, there is continuous evidence that specific staphylococci have inhibitory effects on the proliferation, migration, and other biological behaviors of specific tumors [54,66]. For example, after intratumoral injection of *S. aureus* into the mouse model of orthotopic glioma, delayed glioma growth was observed, which may involve the anti-tumor effect of activated microglia [92].

### 4.1. Surface Adhesion Molecules

As a typical class of adhesion proteins from *S. aureus*, fibronectin-binding protein A/B (FnBPA/B) is associated with the adhesion and costimulatory signals of T lymphocytes [11,12]. The mice which were vaccinated with a recombinant *Lactococcus lactis* stain with cell surface-anchored FnBPA against *S. aureus* were better protected from the human papilloma virus (HPV)-induced cancer [76]. *Aframomum melegueta* extracts the display anti-adhesive abilities of *S. aureus* to lung carcinoma A549 cell line [106]. The extracellular adhesion protein (Eap) of *S. aureus* inhibited the bone metastasis of breast cancer cell line MDA-MB-231 [46]. In addition, some staphylococci were reported to adhere to bladder cancer cells. Szabados, F. et al. observed the internalization of *S. saprophyticus* ATCC 15305 into human urinary bladder carcinoma cell line 5637 in microscopy [49]. The treatment of metabolic glycoengineering with N-azidoacetyl-glucosamine (GlcNAz) leads to the reduced adherence of *S. aureus* to human T24 bladder carcinoma cells [64].

### 4.2. α-hemolysin

The α-hemolysin has certain anti-cancer effects and can also enhance the apoptosis of tumor cells induced by specific chemotherapy drugs [50,68,102]. For instance, a low toxic concentration of α-hemolysin can cause cell apoptosis through the mitochondrial pathway and improve the sensitivity of malignant pleural mesothelioma cells to cisplatin chemotherapy [50]. Additionally, researchers have tried to develop different bacterial delivery systems of α-hemolysin for the targeted killing of colorectal or breast cancer cells using Escherichia coli without the virulence factors [68,102].

### 4.3. Panton-Valentine leukocidin

As the S component of Panton-Valentine leukocidin, LukS-PV can induce mitochondria-mediated apoptosis and G0/G1 cell cycle arrest in human acute myeloid leukemia (AML) cell line (THP-1) [65], and effectively inhibit the tumorigenesis of HL-60 AML cells in severe combined immunodeficiency (SCID) mice [70]. This indicates that LukS-PV may be a multi-target drug candidate for the prevention and treatment of AML. For non-small-cell lung cancer (NSCLC) cells, LukS-PV promotes the apoptosis and cycle arrest of A549 and H460 cells through the P38/ERK MAPK signaling pathway [95]. For liver cancer, LukS-PV inhibits the migration of hepatocellular carcinoma cells by down-regulating histone deacetylase 6 (HDAC6) and increasing α-tubulin acetylation [115], and induces the apoptosis of HepG2 cells by regulating key proteins and metabolic pathways [96].

### 4.4. Staphylococcal Superantigens

Currently, there are many *S. aureus* superantigens, such as SEA, SEB, SEC, TSST1, and SpA, which can exert anti-tumor effects by inducing immune cell death, tumor cell apoptosis and other mechanisms [147,148,149]. Several tumor-specific superantigens for cancer treatment are under development [39,150,151].

#### 4.4.1. Staphylococcus Aureus Enterotoxin A

Enhanced SEA expression in tumor cells with poor immunogenicity increases immunogenicity as a vaccine [53]. In addition, SEA can be utilized in the design of fission superantigen fusion proteins for cancer immunotherapy [41,62,147,151]. For instance, Dohlsten M. et al. designed a C242Fab-SEA fusion protein to target SEA-reactive T cells against MHC-class II negative human colon cancer cells at nanomolar concentrations in vitro [41]. Additionally, an oncolytic adenovirus (PPE3-SEA) was reported to inhibit the growth of mice bladder cancer MB49 cells [62].

#### 4.4.2. Staphylococcus Aureus Enterotoxin B

Like SEA, SEB has significant anti-tumor effects by activating T cells in tumor-bearing mice [38]. Akbari, A. et al. reported that SEB effectively down-regulated the expression of SMAD family members by 2/3 and reduced the proliferation of human primary glioblastoma cell line U87 [80]. Several publications reported the links between SEB and bladder cancer. SEB can activate T lymphocytes and inhibit bladder tumor cell growth in vitro and in vivo [152]. The anti-angiogenic effect of SEB was also observed in an experiment using a rat model of nonmuscle invasive bladder cancer [59]. SEB-stimulated peripheral blood mononuclear cells can lead to the apoptosis of transitional cell carcinoma cells [43]. Similarly, the corresponding modifications of SEB serve as efficient instruments of cancer therapy [60,147]. For instance, Gu L. et al. designed the SEB-H32Q/K173E mutant, which retains the properties of SAg, enhances the host immune response to tumor disease, and reduces the associated thermotoxicity [60].

#### 4.4.3. Staphylococcus Aureus Enterotoxin C

Highly agglutinative staphylococcin (HAS), a mixture of *S. aureus* culture filtrate, plays a certain immunomodulatory role through the active SEC component in the clinical treatment of breast cancer, colon cancer, bladder cancer and other cancers [74,75,153]. As a result, HAS may reduce the side effects of radiotherapy or chemotherapy in specific tumors to a certain extent and improve the survival prognosis of patients [74,153]. In China, SEC2 and a series of mutants have commonly been used as antitumor immunotherapy agents [67,154,155].

#### 4.4.4. Toxic Shock Syndrom Toxin-1

Superantigen TSST-1 was reported to stimulate T-cell activation and enhance the cytotoxic effect of T cells on colorectal cancer LoVo cells [63]. Jiang Y. Q. et al. reported that the fusion of protein TSST-1 with a 12-mer peptide was able to inhibit the hepatocellular carcinoma cell growth by activating T lymphocytes [45]. Additionally, LINC00847 lncRNA may serve as a therapeutic target of the staphylococcal enterotoxin TST gene in renal cell carcinoma [104].

#### 4.4.5. Staphylococcal Protein A

As one of the most essential *S. aureus* cell wall proteins, SpA can be utilized in the clinical treatment of cancer [156]. Based on the cross-linking between SpA and the Fc region of an immunoglobulin, the immunoprecipitation assay of tumor-related protein molecular interactions can be performed, or the delivery system of anti-cancer antibodies or drugs can be prepared [157,158]. For instance, an alkyl vinyl sulfone/protein A-based immunostimulating complex was established to deliver the cancer drugs to trastuzumab-resistant HER2 (human epidermal growth factor receptor 2)-overexpressing breast HCC1954 cells [73].

### 4.5. Others

Other substances of *Staphylococcus* are found to have certain tumor-suppressive effects. First, a protein purified from *Staphylococcus* hominis strain MANF2 was found to have the ability to reduce the viability of colon cancer cell line (HT-29) and lung cancer cell line (A549) when associated with fermented food [103]. Second, the chemotaxis inhibitory protein of *S. aureus* can inhibit the mitochondrial peptide-induced migration of U87 glioblastoma cells [71]. Third, the peptidoglycan of infectious *S. aureus* can actively trigger the Toll-like receptor 2 to promote the invasiveness and adhesiveness of MDA-MB-231 cells in vitro [55]. Fifth, the *S. epidermidis* strain MO34 inhibited the melanoma growth by producing 6-n-hydroxyaminopurine [88,159]. Sixth, cytoplasmic fractions of *Enterococcus faecalis* and *Staphylococcus hominis*, isolated from human breast milk, can inhibit the proliferation of MCF-7 cells [77]. Lastly, *S. aureus*-derived extracellular vesicles enhance the efficacy of tamoxifen therapy in breast cancer cells (MCF7 and BT474) [111].

## 5. Conclusions

The treatment of clinical cancer patients is often complicated with *Staphylococcus* infection, and different tumor treatments are often accompanied by a change in the *Staphylococcus* spectrum. Other types of staphylococci have distinct and even opposite effects on the occurrence and development of specific tumors. Herein, we provided a bidirectional functional effect model of *Staphylococcus* on carcinogenesis, as shown in Figure 1.

To treat cancer patients with bacterial infections, it is important to suppress their complications, starting with the pathogenic mechanism of specific *Staphylococcus*. Targeting the structures, secreted products, or artificial modifications of various virulence factors may result in great success when treating tumors. The accurate and efficient application of specific staphylococcal anti-tumor components also depends on basic experimental evidence, as well as the ongoing improvement of the system for the separation, purification, and presentation of active components.

In this review, we, for the first time, summarize the clinical reports, cellular and animal experimental evidence regarding the association between *Staphylococcus* and the diagnosis and treatment of tumors. Additionally, we systematically investigated the functional links between staphylococci and the occurrence, development, diagnosis, and treatment of breast, skin, oral, colon, and other types of cancers, in terms of surface adhesion molecules, α-hemolysin, PVL, SEs, TSST-1, SpA, and SND1, which provides novel insight into the functional relationship between bacterial infections and tumors.

## Figures and Tables

**Figure 1 microorganisms-10-02353-f001:**
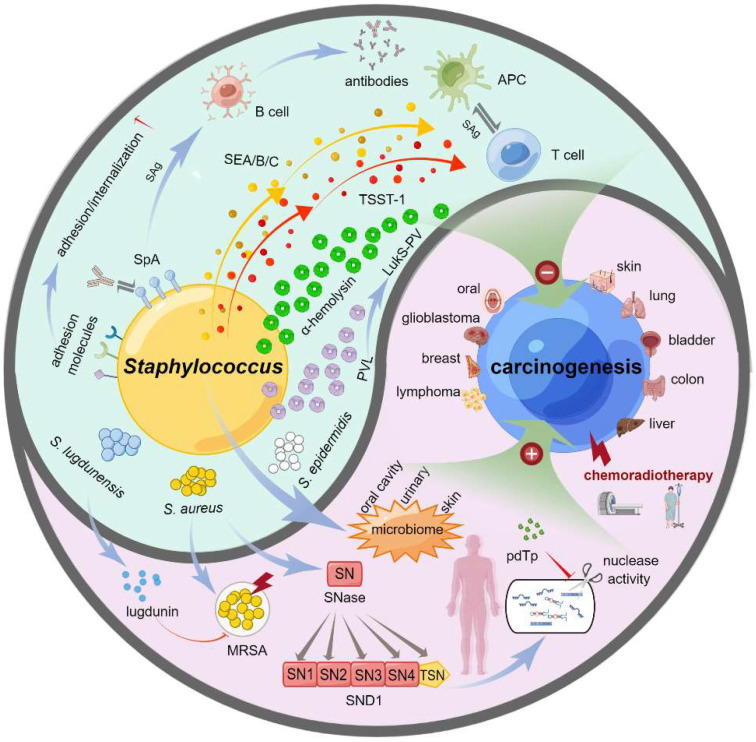
**Bidirectional functional effects of *Staphylococcus* on carcinogenesis**. *Staphylococcus* has the bidirectional effects on carcinogenesis in various types of cancers, such as skin cancer, lung cancer, bladder cancer, colon cancer, liver cancer, lymphoma, breast cancer, glioblastoma, and oral cancer. On the one hand, the changes of staphylococcal flora in some tissues of the body, such as oral cavity, skin or urinary system, was linked to the predisposition to cancer or detected in cancer cases undergoing chemotherapy and/or radiotherapy. MRSA is often associated with a reduced survival rate of patients with malignant tumors. SNases work as the extracellular nucleases of *S. aureus*, and there exists a human homologue of SNases, SND1, which is closely related to the occurrence and development of different cancers. On the other hand, *S. lugdunensis* can secrete a lugdunin to curb the reproduction and infection of MRSA. *S. aureus* may play the role of tumor inhibition through the points of bacterial toxins (alpha-hemolysin, PVL or LurkS-PV), superantigens (SEA/B/C, TSST-1, SpA) of T/B cells, or adhesion molecules. Additionally, the inhibition of SND1 nuclease activity by pdTp may be an effective intervention or therapeutic strategy for liver cancer. This figure was drawn by Figdraw.

**Table 1 microorganisms-10-02353-t001:** Summary of evidence on *Staphylococcus* and carcinogenesis.

Number	Year	Cancer	*Staphylococcus*-Related Issue	Clinical or Experimental Samples	Links	Reference
1	1991	Skin cancer	SEB	PRO4L cell; C3H mice	SEB  V beta 8^+^ cells  tumor growth 	[38]
2	1991	Colon cancer	SEA	SW620, WiDr, COLO205 cells	C215-SEA  anti-tumor 	[39]
3	1992	Several types of cancers	Oral flora	197 patients with advanced malignant disease	*S. aureus* (28% oral rinses)	[40]
4	1995	Colon cancer	C242Fab-SEA	COLO205 cell; humanized SCID mice	C242Fab-SEA  T cell infiltration  tumor growth 	[41]
5	2004	Lung cancer	Tobacco tar-resistant *S. aureus* (Sa-TA10)	H226B cells, Bhas 42	Sa-TA10  TNF-α  carcinogenic potential 	[42]
6	2005	Bladder cancer	SEB	TCC cells	SEB-stimulated PBMC  apoptosis 	[43]
7	2005	Breast cancer	MRSA	One case with ductal breast carcinoma	Complications	[44]
8	2006	HCC	TSST-1	SMMC772 cell	12 mer peptide fused with the TSST-1  migration of tumor cell 	[45]
9	2007	Breast cancer	Eap of *S. aureus*	MDA-MB-231 cell	Eap  bone metastasis 	[46]
10	2007	Several types of cancers	*Staphylococcus*	300 patients with 13 different cancer diagnoses	Frequently isolated *Staphylococcus* during chemotherapy (oral microbiota)	[47]
11	2007	Colon cancer	Tannase	Colon cancer cases vs. adenoma/normal controls (1999~2004)	*S. lugdunensis* (fecal and rectal) 	[48]
12	2008	Bladder cancer	*S. saprophyticus* ATCC 15305	5637 cells	*S. saprophyticus* internalization 	[49]
13	2008	Mesothelioma	α-hemolysin	P31 res cell	α-hemolysin  cytotoxicity 	[50]
14	2008	Glioblastoma	*S. aureus*	One glioblastoma multiforme case	Intracranial abscess complication 	[51]
15	2009	Skin cancer	*S. aureus*	82 skin SCC patients vs. 353 healthy subjects	*S. aureus* DNA (biopsies) 	[52]
16	2009	Melanoma	SEA	B16 cell	SEA-TDLN  pulmonary metastasis 	[53]
17	2009	Several types of cancers	SSL10	Jurkat T-ALL; Jurkat; HeLa cells	SSL10  CXCR4 binding  CXCL12-induced migration of tumor cells 	[54]
18	2010	Breast cancer	Peptidoglycan of *S. aureus*	MDA-MB-231 cell	Peptidoglycan  TLR2  Invasiveness/adhesiveness of tumor cell 	[55]
19	2011	HCC	human homologue of SNases	HepG3, QGY-7703, Hep3B, and Huh7 cells	pdTp  nuclease activity of SND1  RISC activity  hepatocarcinogenesis 	[56]
20	2011	Lung cancer	*S. epidermidis*	32 surgically removed lung cancer samples	*S. epidermidis* 	[57]
21	2012	Oral cancer	*S. aureus* and *S. epidermidis*	186 patients with chemotherapy or chemoradiotherapy (2007~2009)	*S. aureus* and *S. epidermidis* (blood; oral cavity) 	[58]
22	2012	Bladder cancer	SEB	75 female Fisher 344 rats (nonmuscle invasive bladder cancer model)	SEB  anti-angiogenic effects 	[59]
23	2013	Several types of cancers	SEB	BGC823; HeLa cells; mouse Lewis lung carcinoma model	SEB-H32Q/K173E  cytotoxic effects  host immune response 	[60]
24	2013	Cancer MRSA	MRSA	44 cancer cases on therapy vs. 34 non-cancer controls in Saudi Arabia (MRSA isolates)	multiple resistant for antibiotic agents 	[61]
25	2013	Bladder cancer	PPE3-SEA	MB49 cells; mice	PPE3-SEA  CD3^+^ T cells  Tumor growth 	[62]
26	2013	Colorectal cancer	TSST-1	LoVo cell	TSST-1  T cell activation  Cytotoxicity of lymphocytes 	[63]
27	2013	Bladder cancer	*S. aureus*	T24 cell	GlcNAz  adherence 	[64]
28	2013	AML	PVL	THP-1 cell	LukS-PV  apoptosis  cell cycle arrest 	[65]
29	2013	Several types of cancer	egcSEs	Hep-2, CRL5800, CRL1547, MDA-MB-549, SK-N-BE, PLAOD cells	Apoptosis of tumor cells 	[66]
30	2013	HCC	SEC2	Hepa1-6 cell	SEC (14-128)  tumor growth 	[67]
31	2014	Breast cancer	α-hemolysin	MCF7, 4T1 cells, mice	α-hemolysin  necrosis  tumor growth 	[68]
32	2014	Cutaneous T-cell lymphoma	*S. aureus*	Sezary syndrome patients; SeAx, MF1850 cells	*S. aureus* colonization  SEs  Stat3/IL-10 axis  immune dysregulation 	[69]
33	2015	AML	PVL	HL-60 AML cell; SCID mice	LukS-PV  apoptosis  tumor growth 	[70]
34	2015	Glioblastoma	CHIPS	U87 cell; 178 GBM cases	CHIPS  FPR1 activity  U87 migration 	[71]
35	2015	Breast cancer	*S. aureus* and *S. epidermidis*	Cancer patients with breast implantation	*S. aureus* and *S. epidermidis*  breast peri-implant infections 	[72]
36	2016	Breast cancer	SpA	HCC1954 cell	Alkyl vinyl sulfone/protein A complex  cell viability 	[73]
37	2016	Breast cancer	HAS	62 cancer cases	HAS  overall response rate 	[74]
38	2016	Liver cancer	HAS	22 cancer cases	HAS intrahepatic injection  antitumor immune cells 	[75]
39	2016	HPV-induced cancer	FnBPA	Mouse model of HPV-induced cancer	FnBPA  HPV-induced cancer 	[76]
40	2016	Breast cancer	Cytoplasmic fractions of enterococcus faecalis and *Staphylococcus* hominis	MCF-7 cell	Cytoplasmic fractions  proliferation  apoptosis of tumor cell 	[77]
41	2016	Breast cancer	*Staphylococcus*	Women with breast cancer vs. healthy controls	*Staphylococcus* 	[78]
42	2016	BIA-ALCL	Microbiome in breast implant	26 BIA-ALCL samples vs. 62 nontumor capsule specimens	*Staphylococcus* 	[79]
43	2016	Glioblastoma	SEB	U87 cell	SEB  Smad2/3  Proliferation 	[80]
44	2017	Several types of cancers	*S. lugdunensis*; CoNS	Cancer patients with isolated *S. lugdunensis*	*S. lugdunensis* < other CoNS (infection)	[81]
45	2017	Breast cancer	Local breast microbiota	57 Cancer cases vs. 21 negative controls	*Staphylococcus* 	[82]
46	2017	Lung cancer	Lipoteichoic acid of *S. aureus*	A549 and H226 cells	Lipoteichoic acid  proliferation 	[83]
47	2017	HCC	Human homologue of SNases	Hepatocyte-specific SND1 transgenic mice	pdTp  HCC xenografts 	[84]
48	2018	Bladder cancer	Urinary microbiota profile	31 male cancer cases vs. 18 non-neoplastic controls in China	*S. aureus* infection 	[85]
49	2018	Several types of cancers	Oral flora	100 cancer cases vs. 70 healthy controls (oral rinse)	Chemo- and radiotherapy  *S. aureus* counts 	[86]
50	2018	Several types of cancers	Oral microbiota profile	Cancer patients during chemotherapy (17 studies)	Frequently observed *Staphylococcus*	[87]
51	2018	Melanoma	*S. epidermidis* strain MO34	B16F10 cell	MO34  6-n-hydroxyaminopurine  growth of tumor cell 	[88]
52	2018	Colon cancer	*S. lugdunensis*	288 rectal swabs (2002~2008)	Specific group D clone	[89]
53	2019	BIA-ALCL	Microbiota of breast, skin, implant, and capsule	BIA-ALCL and contralateral control breast (*n =* 7)	*Staphylococcus*  (both)	[90]
54	2019	Cancer with MRSA	MRSA	80 HA-MRSA; 40 CA-MRSA isolates from Egyptian cancer patients	Gamma-irradiation  *mecA* gene (HA-MRSA)  multi-antibiotic resistance (CA-MRSA) 	[91]
55	2019	Glioma	*S. aureus*	C57/BL6 mouse model of orthotopic glioma	*S. aureus* intratumoral injection  microglia activation  orthotopic glioma growth 	[92]
56	2020	Cutaneous SCC	*S. aureus*	12 cutaneous SCC cases vs. 28 negative controls, HSC-1 and SCL-1 cells	*S. aureus*  hBD-2  growth of tumor cell 	[93]
57	2020	Lung cancer	*S. aureus*	Cancer patients after lung resection surgery: 108 cases with nasopharyngeal screening vs. 108 controls without screening	*S. aureus* (nasal cavity)  health care-associated infections following lung cancer surgery 	[94]
58	2020	NSCLC	PVL	A549 and H460 cells	LukS-PV  apoptosis  cell cycle arrest 	[95]
59	2020	HCC	PVL	HepG2 cell	LukS-PV  apoptosis  proliferation 	[96]
60	2020	Breast cancer	Breast tumor microbiome	Cancer patients from Black/White non-Hispanic	*Staphylococcus* (second dominant bacterium) 	[97]
61	2020	Breast cancer	Breast microbiota	10 cancer cases vs. 36 healthy controls	*Staphylococcus* 	[98]
62	2020	Breast cancer	Breast tumor microbiome	Cancer cases with distant metastases vs. cancer cases without metastases	*Staphylococcus* 	[99]
63	2020	Several types of cancers	SAB	SAB cohort (*n* = 12,918); Population cohort (*n* = 117,465)	SAB  risk of primary cancers 	[100]
64	2020	Breast cancer	*S. aureus*	4T1 cell	*S. aureus* infection  NET  Lung metastasis 	[101]
65	2020	Colorectal cancer	α-hemolysin of *S. aureus*	SW480 cell	Light-activated recombinantα-hemolysin  Apoptosis or necrosis of tumor cell 	[102]
66	2020	Colon/lung cancer	*Staphylococcus* hominis strain MANF2	A549 and HT-29 cells	MANF2  Viability of tumor cells 	[103]
67	2020	RCC	TSST-1	ACHN cell	*tst* gene  LINC00847  apoptosis 	[104]
68	2021	Glioblastoma	*Staphylococcus*	29 glioblastoma cases with cerebral infections (four studies)	Staphylococcal intracranial infection  longer survival time  (in one study)	[105]
69	2021	Lung cancer	*S. aureus* (ATCC 29213)	A549 cells	*Aframomum melegueta* extract  Adhesion of *S. aureus* to A549 	[106]
70	2021	Breast cancer	*Staphylococcus*	221 cancer cases vs. 69 negative controls	*Staphylococcus* 	[107]
71	2021	Bladder cancer	Bladder microbiota	Tumor mucosa samples of 32 patients (2010~2017)	*Staphylococcus* (cluster 2) 	[108]
72	2021	Several types of cancers	MRSA	Patients with malignancy (2000–2020)	MRSA BSIs  mortality rate 	[109]
73	2022	Oral cancer	Microbiota profile	27 oral cancer cases vs. 15 healthy subjects	*Staphylococcus* 	[110]
74	2022	Breast cancer	*Staphylococcus*; *S. aureus* derived EVs	96 cancer cases vs. 192 healthy controls; MCF7 and BT474 cells	*Staphylococcus*  EVs  Endocrine therapy efficacy of tumor cells 	[111]
75	2022	prostate cancer	Urinary microbiota	50 cancer cases undergoing radiotherapy	*S. haemolyticus*; *S. epidermidis*; *S. hominis* 	[112]
76	2022	Several types of cancers	Bacterial profile and antimicrobial susceptibility	200 cancer cases (2021.03–2021.07)	*S. aureus* (51.5%)	[113]
77	2022	Bladder cancer	*Staphylococcus* level	Bladder cancer vs. Benign Prostatic Hyperplasia	*Staphylococcus* (urine) 	[114]
78	2022	HCC	PVL	HepG2, Bel-7402, Hep3B, Huh-7 cells	LukS-PV  HDAC6  α-tubulin acetylation  migration 	[115]


 upregulation or enhancement; 

 downregulation or reduction; vs.:versus; SEB: staphylococcal aureus enterotoxin B; SEA: staphylococcal aureus enterotoxin A; SCID: severe combined immunodeficiency; TNF-α: tumor necrosis factor-α; TCC: transitional cell carcinoma; PBMC: peripheral blood mononuclear cells; MRSA: methicillin-resistant *Staphylococcus aureus*; HCC: Hepatocellular carcinoma; TSST-1: toxic shock syndrome toxin-1; Eap: extracellular adhesion protein; SCC: squamous cell carcinoma; TDLN: tumor-draining lymph nodes; SSL10: staphylococcal superantigen-like 10; CXCR4: C-X-C motif chemokine receptor 4; CXCL12: C-X-C motif chemokine ligand 12; TLR2: Toll-like receptor 2; pdTp: 3′,5′-deoxythymidine bisphosphate; SND1: staphylococcal nuclease and Tudor domain-containing 1; GlcNAz: N-azidoacetyl-glucosamine; AML: acute myeloid leukemia; PVL: Panton-Valentine leukocidin; egcSEs: staphylococcal entertotoxins of the enterotoxin gene cluster; SEC2: staphylococcal aureus enterotoxin C2; SEs: staphylococcal enterotoxins; CHIPS; chemotaxis inhibitory protein of *S. aureus*; FPR1: Formyl peptide receptor 1; SpA: staphylococcal protein A; HAS: highly agglutinative staphylococcin; HPV: human papilloma virus; FnBPA: fibronectin-binding protein A; BIA-ALCL: breast implant-associated anaplastic large-cell lymphoma; Smad2/3: SMAD family member 2/3; CoNS: coagulase negative staphylococci; HA-MRSA: hospital-acquired MRSA; CA-MRSA: community-acquired MRSA; HBD-2: β-defensin-2; CRC: Colorectal cancer; NSCLC: Non-small-cell lung cancer; SAB: *S. aureus* bacteremia; NET: neutrophil extracellular traps; RCC: renal cell carcinoma; ACHN: human renal cell adenocarcinoma; BSI: bloodstream infection; Vs: extracellular vesicles; HDAC6: histone deacetylase 6.

## Data Availability

Availability of published literature and correspondence should be addressed to the corresponding author.

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
