# Peer review of "Bidirectional Functional Effects of Staphylococcus on Carcinogenesis"

_microorganisms, 2022, doi:10.3390/microorganisms10122353_

Round 1
Reviewer 1 Report
Overall, the manuscript is correctly written. The current manuscript is interesting and novel, so due to that the manuscript could of valuable contribution to the scientific community. So, generally, the study appear to be sound, well-designed and could be also of practical value. The use of English is mostly very good. Therefore, the undertaken issue is interesting and worth of disseminating. However, I suggest that authors add several recent references in order to additionally improve the manuscript.
References which I suggest in the manuscript are following:
Cvetnić, L., M. Samardžija, S. Duvnjak, B. Habrun, M. Cvetnić, V. Jaki Tkalec, D. Đuričić, M. Benić (2021): Multi Locus Sequence Typing and spa Typing of Staphylococcus aureus Isolated from the Milk of Cows with Subclinical Mastitis in Croatia. Microorganisms 9, 725. doi.10.3390/microorganisms9040725
Saidi, R., Z. Cantekin, N. Mimoune, Y. Ergun, H. Solmaz, D. Khelef, R. Kaidi (2021): Investigation of the presence of slime production, VanA gene and antiseptic resistance genes in Staphylococci isolated from bovine mastitis in Algeria. Vet. stn. 52, 57-63. doi.org/10.46419/vs.52.1.9
Author Response
Reviewer 1
Overall, the manuscript is correctly written. The current manuscript is interesting and novel, so due to that the manuscript could of valuable contribution to the scientific community. So, generally, the study appears to be sound, well-designed and could be also of practical value. The use of English is mostly very good. Therefore, the undertaken issue is interesting and worth of disseminating. However, I suggest that authors add several recent references in order to additionally improve the manuscript.
References which I suggest in the manuscript are following:
Cvetnić, L., M. Samardžija, S. Duvnjak, B. Habrun, M. Cvetnić, V. Jaki Tkalec, D. Đuričić, M. Benić (2021): Multi Locus Sequence Typing and spa Typing of Staphylococcus aureus Isolated from the Milk of Cows with Subclinical Mastitis in Croatia. Microorganisms 9, 725. doi.10.3390/microorganisms9040725
Saidi, R., Z. Cantekin, N. Mimoune, Y. Ergun, H. Solmaz, D. Khelef, R. Kaidi (2021): Investigation of the presence of slime production, VanA gene and antiseptic resistance genes in Staphylococci isolated from bovine mastitis in Algeria. Vet. stn. 52, 57-63. doi.org/10.46419/vs.52.1.9
Response: Thanks for your nice comment. Based on your useful suggestion, we have added the two references in the introduction section [reference 1, 5]. Please check the lines 38-42, page 2.

Reviewer 2 Report
The submitted review manuscript gathers scientific data on the links between the presence of Staphylococcus and the occurrence, development, and treatment of different types of cancer. This kind of work significantly contributes to the field and can be found interesting for the scientific community working both on cancer and on microbiomes of patients and healthy individuals. The main drawback of the review is the lack of a description of how the literature data has been searched and selected (comment to line 80).
Remarks to the manuscript
The information about Table 1 should be at the beginning of the manuscript, not only in the Conclusion section.
Latin bacterial like genus (Staphylococcus) and species (e.g. Staphylococcus aureus, S. epidermidis, E. coli) names should be in italics. English words derived from Latin names (e.g. staphylococcal) should be in normal font and start with a small letter.
Line 20 “a variety of types” do you mean species? (you mention them later)
Line 24 “to normal controls” do you mean “healthy controls”?
Line 30 “Interestingly, the human homologue of SNases is SND1 (Staphylococcal nuclease and Tudor domain containing 1), which is recognized as an oncoprotein.” Please reformulate, as it is not clear. Do you maybe mean “Interestingly, the human homologue of SNases is SND1 (Staphylococcal nuclease and Tudor domain containing 1) is recognized as an oncoprotein.”
Line 38 “cocci that”
Line 52 “Various cell wall-protein anchored surface proteins”
Line 80 Please provide which databases and keywords were used for article searching and how they were selected
Line 102 ” a relative abundance” please specify this expression by giving some numbers (range, percentage etc.)
Line 169 “hydrophobic”
Line 191 – please explain the abbreviation RISC at the first use
Line 222 “For instance, interfering with oxidative stress in staphylococci and tumors through a variety of strategies may help to achieve effective treatment and drug resistance against bacteria and cancer from the point of reactive oxygen species” please reformulate as it is not clear.
Line 231 “fibronectin-binding”
Heading of 4.3, 4.4 should be full names not abbreviations
Line 297 “An experiment was conducted by Jiang Y. Q. et al. where TSST-1 was fused with a 12mer peptide to target HCC cells and inhibit tumor growth [147].” And what was the effect of it?
Line 318 “Enterococcus faecalis and Staphylococcus hominis ”
Line 329 do you mean Figure1?
Table 1. Please correct “microflora” to “microbiota” (row 10 and 75)
Author Response
Reviewer 2
The submitted review manuscript gathers scientific data on the links between the presence of Staphylococcus and the occurrence, development, and treatment of different types of cancer. This kind of work significantly contributes to the field and can be found interesting for the scientific community working both on cancer and on microbiomes of patients and healthy individuals. The main drawback of the review is the lack of a description of how the literature data has been searched and selected (comment to line 80).
Response: Thanks for your professional comment. Based on your useful suggestion, we have added the necessary information for the reference search and selection process. Please check the lines 80-99 (page 3-4).
Remarks to the manuscript
The information about Table 1 should be at the beginning of the manuscript, not only in the Conclusion section.
Response: Thanks for your professional comment. Based on your useful suggestion, we have placed the Table 1 in the introduction section. Please check the lines 88-90 (page 4).
Latin bacterial like genus (Staphylococcus) and species (e.g. Staphylococcus aureus, S. epidermidis, E. coli) names should be in italics. English words derived from Latin names (e.g. staphylococcal) should be in normal font and start with a small letter.
Response: Thanks for your professional comment. Based on your useful suggestion, we have revised the presentation of these nouns in the whole text, Table and Figure. Please check our revised version.
Line 20 “a variety of types” do you mean species? (you mention them later)
Response: Thanks for your nice comment. It should be species. We have revised this point.
Line 24 “to normal controls” do you mean “healthy controls”?
Response: Thanks for your nice comment. We have revised this point, based on your useful suggestion.
Line 30 “Interestingly, the human homologue of SNases is SND1 (Staphylococcal nuclease and Tudor domain containing 1), which is recognized as an oncoprotein.” Please reformulate, as it is not clear. Do you maybe mean “Interestingly, the human homologue of SNases is SND1 (Staphylococcal nuclease and Tudor domain containing 1) is recognized as an oncoprotein.”
Response: Thanks for your nice comment. We have revised this sentence, based on your useful suggestion. “Interestingly, a human homologue of SNases, SND1 (staphylococcal nuclease and Tudor domain containing 1), has been recognized as an oncoprotein”. Thanks.
Line 38 “cocci that”
Line 52 “Various cell wall-protein anchored surface proteins”
Response: Thanks for your nice comment. We have revised the two points.
Line 80 Please provide which databases and keywords were used for article searching and how they were selected
Response: Just as stated above, we have added the necessary information in the lines 88-90 (page 4)..
Line 102 ” a relative abundance” please specify this expression by giving some numbers (range, percentage etc.)
Response: Thanks for your nice comment. We carefully checked the relative references for this sentence. Some references did not provide the detailed number information. Thus, we illustrate "a relative abundance" by taking examples. Namely. “For instance, as the second dominant bacterium, Staphylococcus (6.4% ± 9.4%) was prevalent in 22 out of 23 breast tissue samples of cases within Black or white non-Hispanic cohorts of breast cancer (Thyagarajan et al., 2020)” Please chec it. Thanks, again.
Line 169 “hydrophobic”
Response: Thanks for your nice comment. We have revised this point.
Line 191 – please explain the abbreviation RISC at the first use
Response: Thanks for your nice comment. We have revised this point. Please check lines 69-72, page 3.
Line 222 “For instance, interfering with oxidative stress in staphylococci and tumors through a variety of strategies may help to achieve effective treatment and drug resistance against bacteria and cancer from the point of reactive oxygen species” please reformulate as it is not clear.
Response: Thanks for your nice comment. We have revised this sentence. That is “For instance, it may be possible to evade drug resistance in Staphylococcus and tumors by regulating intracellular reactive oxygen species.”
Line 231 “fibronectin-binding”
Response: Thanks for your nice comment. We have revised this point.
Heading of 4.3, 4.4 should be full names not abbreviations
Response: Thanks for your nice comment. We have revised this point.
Line 297 “An experiment was conducted by Jiang Y. Q. et al. where TSST-1 was fused with a 12mer peptide to target HCC cells and inhibit tumor growth [147].” And what was the effect of it?
Response: Thanks for your nice comment. We have revised this point. “Jiang Y. Q. et al. reported that the fusion protein of TSST-1 with a 12-mer peptide was able to inhibit the hepatocellular carcinoma cell growth by activating T lymphocytes.”
Line 318 “Enterococcus faecalis and Staphylococcus hominis ”
Response: Thanks for your nice comment. We have revised this point.
Line 329 do you mean Figure1?
Response: We are very sorry for our mistake. We have corrected this point.
Table 1. Please correct “microflora” to “microbiota” (row 10 and 75)
Response: Thanks for your nice comment. We have corrected this point.
